# Site conditions for regeneration of climax species, the key for restoring moist deciduous tropical forest in Southern Vietnam

**Ha T. T. Do**[1,2]*, **John C. Grant**[1], **Heidi C. Zimmer**[1], **Bon N. Trinh**[2], **J. Doland Nichols**[1]

**1** Forest Research Centre, Southern Cross University, Australia, **2** Silviculture Research Institute, Vietnam Academy of Forest Sciences, Ha Noi, Vietnam

* t.do.16@student.scu.edu.au

**Data Availability Statement:** All relevant data are within the paper and its Supporting Information files.

**Funding:** This study was undertaken as part of HD's PhD research program. HD's PhD is

## Abstract

Understanding the requirements and tolerances of the seedlings of climax species is fundamental for tropical forest restoration. This study investigates how the presence and abundance of seedlings of a previously dominant, now threatened species (*Dipterocapus dyeri* Pierre), varies across a range of environmental conditions. *Dipterocapus dyeri* seedling abundance and site characteristics were recorded at 122 observation points (4 m$^2$) at nine clusters from two sites. Seedling presence (p = 0.065) and abundance varied significantly (p = 0.001) between the two sites, and was strongly correlated with adult *D. dyeri* dominance and lower soil pH, and weakly correlated with canopy openness and total stand basal area. *Dipterocarpus dyeri* seedlings were also grown in shade houses with three light levels on two soils. Seedling survival was significantly lower at the lowest light level (<10% full irradiance) at 13% for the forest soil and 25% for degraded soil. At higher irradiance the seedling survival rates were greater than 99%. Moisture levels remained high at the lowest light level and many seedlings died from fungal infection. We concluded that secondary forests which contain adequate numbers of adult *D. dyeri* as seed sources, light availability, soil pH of < 5.0, and good drainage strongly favour survival and growth of *D. dyeri* seedlings. Historically, *D. dyeri* was dominant in moist deciduous tropical forest across south-eastern Vietnam, but today it is rare. Active management of these recovering forests is essential in order to recover this high-value, climax forest species.

## 1. Introduction

Water, nutrients and light are key elements in plant growth and survival. These resources vary spatially, often with topographic gradients, and temporally, including as they interact with vegetation development. Water and nutrient availability in the soil is largely driven by landscape-scale factors, such as topography, substrate and climate [1,2]. Light availability on the forest floor is largely determined by the density of the forest canopy (i.e., local-scale factors), which changes through time as the forest develops [3,4]. Competition for light is a major influence on recruitment in old-growth tropical forest [5]. However, in secondary tropical forest,

supported by Vietnam Government Ministry of
Education and Training 'Vietnam International
Education Development Program' scholarship
(subsidence living expenses). SCU Forest Research
Centre has provided a scholarship for HD's
university fees. HD's research occurred within the
broader project ĐTĐL.XH.10/15, which is funded
by the Vietnam Government Ministry of Science
and Technology at DNBR. The funders had no role
in study design, data collection and analysis,
decision to publish, or preparation of the
manuscript.

**Competing interests:** The authors have declared
that no competing interests exist.

competition for water is more important than competition for light and drought tolerance is
critical for survival [6–8]. This is because there is often more light reaching the forest floor,
due to gaps in the canopy. Gap size directly influences light availability at the soil surface,
which not only affects the germination and growth of plants in the understory, but also leads
to changes in underground processes, potentially influencing soil temperature, water and tree
root competition [9]–higher light and temperature at the soil surface increasing the likelihood
of drought [10]. For example, single tree fall gaps can expose high-nutrient soil ideal for germi-
nation [11,12].

The mosaic distribution patterns of tree populations and tree ages in tropical forests are the
results of species-specific and tree age-specific adaptations to the spatial distributions of water
and nutrients [13–16]. Soil type, topography and vegetation structure interact to influence
seedling establishment spatially and temporally [17–19]. The structure and composition of a
vegetation community are maintained temporally by species' successful regeneration [11]. Dif-
ferent species have different environmental requirements and different levels of tolerance to
the stresses presented by different environments/site-conditions. For example, low soil pH is
tolerated by many conifer species [20]. The optimum range for growing most pine seedlings is
around 4.5 to 5.0 for sandy soils and 5.0 to 5.5 for fine-texture soils containing high levels of
Mn [21]. *Wollemia nobilis* seedlings, for example, have an optimum soil pH of around 4.3 [22].
*Wollemia nobilis*, like many other conifers, thrives at light levels higher than typically occurs in
most rainforest understories. In contrast, many broadleaf tropical species seedlings such as
*Neolitsea obtusifolia*, *Hopea odorata* and *Brosimum alicastrum* survive better under conditions
those low light levels [23–26]. In order for recruitment to be successful, seeds need to fall
where light and soil moisture conditions are favourable for germination and growth [6,27]. In
some cases, the original conditions that have led to occupation by a specific vegetation may
change to be no longer suitable for its regeneration. The regeneration of a species in tropical
forest can therefore be discontinuous, with not all life stages of all species present. Mimicking
these natural processes, close-to-natural silviculture techniques and nurse crop establishment
were used commonly in the tropical areas as the foundational practise for middle- or late-suc-
cessional afforestation during their seedling and sapling periods. Then light liberation is
applied as seedling increase in height, depending on specific species light requirements [28–
31].

The key to regeneration may be related to the availability of appropriate microsites i.e.,
small-scale variation in resources and species-specific effects [32,33]. When forest stand struc-
ture influences the distribution of water, nutrient and light at the local scales, the species com-
position showed local ecological intra and inter species effects provided a strong influence on
seedling growth [32]. Conspecific density (also known as congeneric) affects not only seed
availability [34–36], but also physically and biologically affects seed germination, survival and
growth of seedlings through negative-density dependence mechanisms [37–39]. Negative-den-
sity dependence mechanisms were shown to strongly negatively influence same species in
monoculture plantation [37,40,41]. In mixed species natural forests, negative-density depen-
dence trends (from positive to negative) with strength of the negative-density dependence
mechanism on specific-species recruitment strongly depended on the quality and quantity of
adult trees in the site (i.e., density, distribution and size of remnant trees).

Dipterocarpaceae was previously abundant throughout southeast Asia, however most
remaining forests are heavily disturbed and are dominated by deciduous tree species and bam-
boo [42–44]. Dipterocarp species are known to be strongly associated with specific soil and
light conditions, especially as seedlings. Microsite light availability, determined by gap size, is
likely to be the most influential factor in natural regeneration of dipterocarps, as dipterocarp
species' light requirements change with life stage. They prefer shade for germination, partial

light during seedling and sapling stages, and full light as mature trees [45–47]. The effect of light availability appears to be stronger than the effect of foliar herbivory on Dipterocarp seedling survival and growth [48,49]. Understanding patterns of natural regeneration of dipterocarps will contribute to efforts to restore and re-establish this highly valued and important type of forest [50–52].

*Dipterocarpus dyeri* is native to Myanmar, Thailand, Viet Nam, Cambodia and Peninsular Malaysia. The species has been globally assessed as Endangered in the IUCN Red list [53,54]. Most dipterocarp species, including *Dipterocarpus dyeri* Pierre, are high value timber species, and as such, have been selectively logged across South-east Asia [42]. *Dipterocarpus dyeri* has been targeted in particular because of its high density wood ($\sim 0.8$ g/cm$^3$) and also their resins (dammar) which can be extracted for caulking boats, varnish paint, and medicine [42,54–57]. *Dipterocarpus dyeri* today is one of the rarest species in the forest and there is limited information on species ecology regeneration and growing site conditions [54,58].

Seedling recruitment is a crucial process in long term forest dynamics. An accurate understanding of tree recruitment patterns in forest stands and microhabitats is required to predict forest development and optimise forest management to achieve ecological and/or production goals [59–61]. This study aimed to determine the primary ecological processes driving *Dipterocarpus dyeri* seedling establishment and survival. The micro-scale conditions belong to three main groups: (1) soil, including particle size distribution and chemical properties and encompassed in soil type; (2) site, including topographic wetness; and (3) forest stand characteristics, including canopy openness, basal area, species biodiversity and dominance of adult *D. dyeri*. Determining the requirements for *D. dyeri* seedlings will improve understanding of secondary forest regeneration. This will help land managers to choose suitable sites to manage regeneration, prepare and plant nursery stock, and develop appropriate silviculture techniques for restoration of dipterocarp forests in southeast Vietnam.

## 2. Methods

### 2.1. Study sites

The study was conducted in a monsoonal evergreen broad-leaved forest in the Dong Nai Biosphere Reserve (DNBR), Dong Nai Province, Vietnam (Fig 1; the maps were sourced from Natural Earth [62]), which are licensed under a CC BY-SA 4.0.). The Director of DNBR Dr. Tran Van Mui, gave permission to conduct this research project in DNBR. The Reserve extends between 11˚20'50"N– 11˚50'20"N and 107˚09'05"E– 107˚35'20"E [63]. The area is in the lowland monsoon tropical climate zone with mean annual rainfall of 1,850 mm of which 80% falls in the wet season (from April to October). The mean annual temperature is approximately 26˚C, with small seasonal fluctuations [64].

Two study sites were established with elevations that ranged from 44 to 130 m asl. A mixed-bamboo (MB) site (UTM 48N 736532E 1257916N (Plot SK3); > 100 m elevation), at Phu Ly Commune was in a steep landscape (10-20˚) and a mixed-evergreen (ME) site (UTM 48N 716275E 1236117N (Plot DD2); 70–95 m elevation) occurred on gentle slopes.

The soils at both sites were classified as Acrisols (grey podzolic soils) [65]. The ME site soils are mapped as Chromic Acrisols (Fp), developed on old alluvium. These soils are yellowish brown and clayey throughout with a Bts horizon at 30–100 cm depth. The MB site soils are Endolithic Chromic Acrisols (Fs), which had developed on schist and shales [65]. The soils in this site are shallow (50–70 cm), yellowish brown sandy clay loam to light clay textures and a BCts horizon at 30–40 cm above R or C horizons. The soil chemical analysis (S1 Table) showed a high level of base saturation (more than 50%), indicating both soils might be better classified as Luvisols [66].

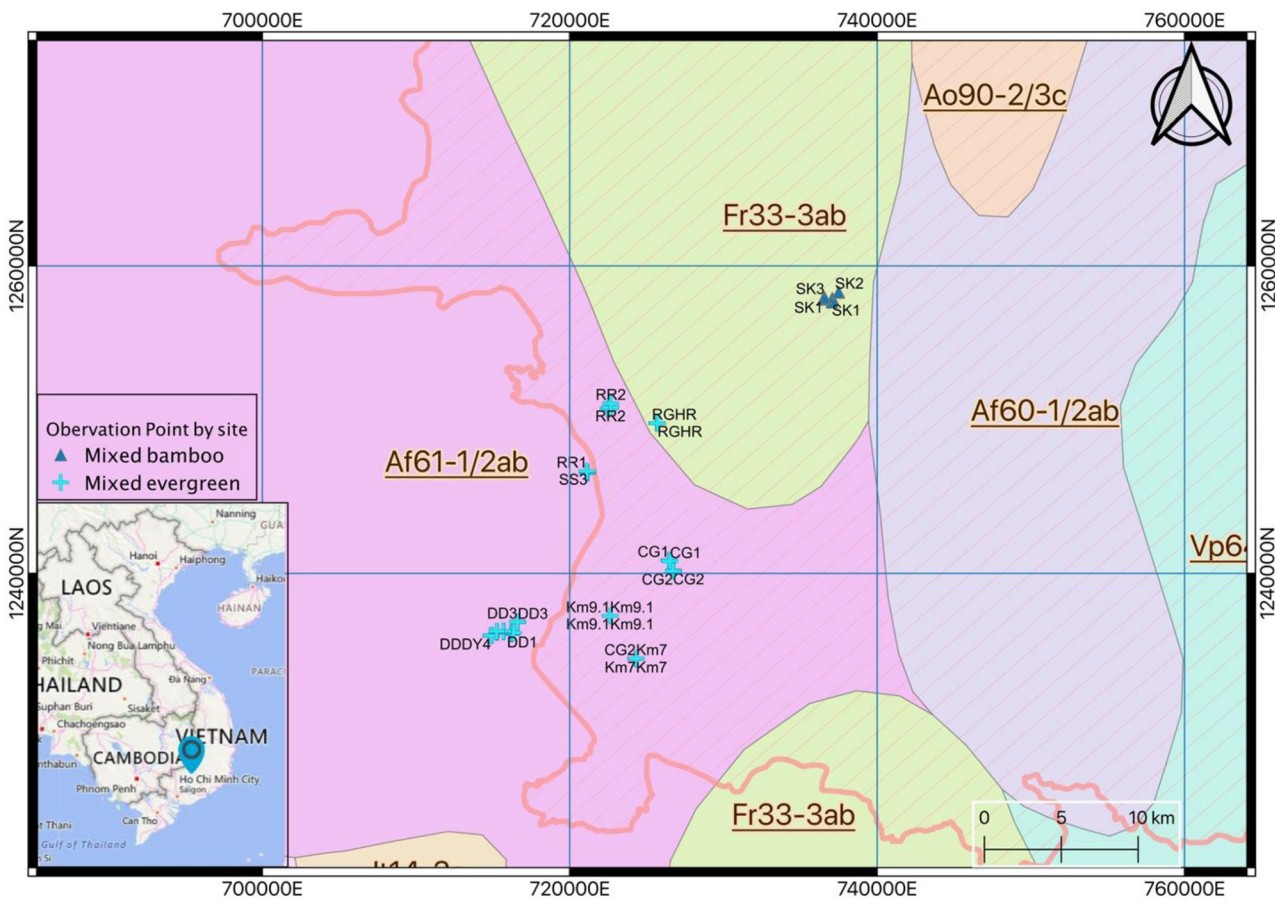

**Fig 1. Location of study site and plots in Dong Nai Province, Vietnam.** Colours on the basemap represent forested (green) and unforested (yellow and brown) land cover. The land cover basemap, country and province borders were sourced from NaturalEarth (http://www.naturalearthdata.com/about/terms-of-use/).

The forests in the study area are lowland tropical forests dominated by dipterocarp and *Lagerstroemia* (Lythraceae) species [67–69]. Most of forests in the area are recovered after logging (early 1943 and from 1976 to 1996), herbicide spraying (1961–1971) and other deforestation activities (1976–1996) [70,71]. The forests at the ME site contained mixed evergreen tree species (*D. dyeri*, *Hopea recopei*, *Vatica odorata*, *Artocarpus* spp., *Syzygium* spp.) and deciduous species (*Lagerstroemia* spp., *Cratoxylum* spp., *Nephelium melliferum)*. The MB site was dominated by bamboos which were considered as a regrowth stage of broadleaf forest after disturbances [68,69].

## 2.2. Study species

*Dipterocarpus dyeri* is a large lowland evergreen rainforest tree species. It also occurs in dry evergreen forests as a brevideciduous species, as they lose their leaves for only a few days [42,72]. In DNBR, according to our observations since 2016, fruiting of adult *D. dyeri* trees (DBH more than 40 cm) occurs almost every year with flowering in November–December and fruit maturation in March–April.

## 2.3. Data collection

The data from the forests were collected during the dry season in November 2017 –January 2018 inside the DNBR (Fig 1). A total of 122 observation points (2 m x 2 m) were taken from 18 sampling clusters (represented by plot vegetation data, Table 1, Fig 1, S1 Data). The locations of these clusters and points were allocated on two sites with 101 points (15 plots) on Mixed Evergreen (ME), and 21 points (3 clusters) on Mixed Bamboo (MB), approximately 20 km from ME site (Fig 1). First, we identified potential *D. dyeri* (DBH > 40 cm) mother trees. Then a search for *D. dyeri* seedlings was undertaken with distance from the mother tree 100 m. If a seedling was found, then that seedling became a corner of a 400 m$^2$ plot. Then all seedlings within that area were identified and measured. If no seedlings were found, then a similar 400 m$^2$ plot was established for surveying site and soil conditions, with a mother tree as the centre and five observation points (each of the four corners and at the centre of the plot).

**2.3.1 Vegetation composition.** In each plot, all woody species (excluding lianas) were recorded and divided into three classes based on the diameter at 1.3 m height from the ground (DBH): adults (DBH ≥ 10 cm), juveniles (5 cm ≤ DBH < 10 cm). Adults and juveniles were surveyed from 400 m$^2$ plots while seedlings of all species were surveyed from the observation points.

**2.3.2 Soil.** At each observation point, a description of soil from 0–35 cm depth was made, including horizons, colour and texture. One topsoil (0–10 cm depth) sample which was a mixture of soil from 3–5 observation points was taken for soil chemical characterisation of each

**Table 1. Description of soil and site characteristics from 122 observation points.**

| Variable | Abbreviation | Mixed Evergreen | Mixed Bamboo |
|---|---|---|---|
| Cluster | Plot | 15 | 3 |
| Observation points (presence) | | 101 (67) | 21 (9) |
| Elevation (m) | Elevation | 83 (13) | 137 (26)[a] |
| Parent materials | Parent.m | Alluvium | Schist and Shale |
| Slope (Degrees) | Slope | 8.6 (7.4) | 11.7 (7.5)[a] |
| Stand basal area (m$^2$/Ha) | BA | 35.3 (12.6)[a] | 17.5 (16.6) |
| Dominance of adult *D. dyeri* | PD.A | 0.27 (0.18)[a] | 0.22 (0.08) |
| Count of D. dyeri seedlings | dyeri | 1.1 (1.6)[a] | 0.4 (0.5) |
| Canopy openness (%) | CO | 12.9 (3.5) | 16.3 (5.1)[a] |
| Topographic Wetness Index | TWI | 11.1 (2.03) | 10.3 (1.33) |
| Topsoil bulk density (g/cm$^3$) | Topsoil.BD | 1.13 (0.21) | 1.11 (0.04) |
| Topsoil pH (KCl) | pH | 3.8 (0.2) | 3.7 (0.1) |
| Topsoil organic carbon (%) | OC | 2.6 (1.1) | 1.97 (0.2) |
| Topsoil total nitrogen (%) | N | 0.23 (0.06) | 0.21 (0.02) |
| Carbon: Nitrogen ratio | CN | 10.7 (1.98)[a] | 9.3 (0.3) |
| Total phosphorous (mg/100g) | P$_2$O$_5$ | 7.6 (2.8)[a] | 5.2 (0.15) |
| Cation Exchange Capacity | CEC | 8.5 (2.6) | 8.9 (0.9) |
| Sand (0.2–2 mm) (%) | Sand | 24 (12)[a] | 17 (8) |
| Fine Sand (0.05–0.2 mm) | Fine.Sand | 18 (6) | 20 (7)[a] |
| Silt (0.002–0.05 mm) | Silt | 30 (12) | 31 (7) |
| Clay (<0.002 mm) (%) | Clay | 26 (13) | 32 (6)[a] |
| Fine particles Silt + Clay (%) | Silt.clay | 57 (14) | 63 (4) |

Mean (SE)

[a] Significant different between two site with wilcox.test function significant level 0.05 [79]

sampling cluster. At each sampling cluster one core sample (5 cm diameter x 5 cm length) was taken for topsoil bulk density calculation.

**2.3.3 Light.** A hemispherical canopy photograph was taken at each observation point at 170 cm above the ground. We chose this height to avoid the shrub and herb layers. These photos were used to determine canopy openness (%) using Gap Light Analyzer (GLA) software [73], after processing the image through SideLook [74] using automatic thresholding. The procedure of using automatic threshold to adjudge colour canopy photographs is recommended for assessing canopy openness [75].

**2.3.4 Environment.** Basic site conditions were described at each observation point, including slope (degrees), aspect (degrees), outcrop rock (% cover), forest floor vegetation cover (%), litter cover (%) and litter thickness (cm). Topographic wetness index (TWI) [76], was calculated as a soil moisture indicator [77,78] for each observation point was derived by Arctoolbox from a digital elevation model (DEM) through SAGAGIS, AcrGIS v 10.1. A DEM of 30 x 30 m grid spacing was downloaded from https://earthexplorer.usgs.gov/.

**2.3.5 Shade house experiment establishment.** Seeds were collected from one mother tree from Cay Gui Forest Ranger Station (approximately 500 m from plot CG1). The seedlings were germinated using moist sand for 20–30 days until developing to seedlings with two true leaves (S2A Fig) which were then transplanted in plastic pots (8 cm diameter x 40 cm height). There were six treatments of three light levels (high, medium and low) and two sources of soil (old growth forest and heavily logged and degraded forest) with a total of 432 seedlings (72 seedlings per treatment). Each light treatment was set up in a shelter with light levels controlled by shade cloth, one layer for the high level, doubling for the medium level and tripling for the low level. The light levels were confirmed by the measurement of leaf area index (LAI) by using Li-2200. The LAI at the three shelters were 2.9, 5.4 and 7.1 giving estimations of canopy openness for each light levels were proximately 45%, 20% and 5% of full sun, respectively. Soil used for the trial was the topsoil of a Chromic Acrisol (Fp) and it was collected from two sites. Both soils were light clays in texture and the old growth forest soil (F) was higher in organic matter, total nitrogen and CEC than the degraded site soil (D) (S1 Table).

## 2.4. Data analysis

**2.4.1. Forest data: Correlation between presence/number of *D. dyeri* seedlings and microsite conditions.** The effect of site (mixed evergreen and mixed bamboo sites) on the assemblage of seedlings of the target species per observation point was tested using multivariate models [80] through manyglm function in mvabund R package [81]. This function analyses the individual and interactive influences of each site condition variable based on mixed effect models. Negative binomial family of distribution link was chosen as seedling abundance data was overdispersed.

The sets of variables were also used to test their effects on presence and abundance of *D. dyeri* seedlings at observation points in generalized linear regression models (GLM). We used the glm.nb function within 'MASS' [82] to fit models with a negative binomial family with distribution for data collected at observation points where seedlings occurred. We also used function glm from package 'lme4' [83] with binomial family for presence/absence data. In the GLM, the cluster (plot) of observation point was not treated as random factor since the forest stand variables were aggregated in the cluster. The final models were selected by AIC in the stepwise search through function step in stats R package stats [79].

As canopy openness was one of the factors of most interest in the forests, we used Bayes t. test function in BayesianFirstAid R package [84] to compare the difference in canopy openness

against seedling abundance level groups and between presence and absence groups for all observation points and for each site.

**2.4.2. *D. dyeri* seedling in shade houses: Influence of soil and light.** To investigate the differences in seedling survival according to light level and soil we used the Pearson Chi-squared test. Analysis of variance (ANOVA) was used to compare the differences in seedling growth (based on diameter and height) among treatments (soil types and light levels, and their interactions).

# 3. Results

## 3.1. Forest data: Differences in seedling abundance in varying soil and site conditions

Correlations between soil and stand properties differed between the two sites (S1 Fig). Stand basal area (BA) was weakly correlated with canopy openness (CO) on the mixed evergreen (ME) site and moderately correlated with proportion of adult *D. dyeri* on the higher elevation mixed bamboo (MB) site. Canopy openness showed weak negative correlations with soil CN, $P_2O_5$ and sand content, and these correlations were similar on the higher site (S1 Fig). For the observation points where the seedlings were present, the correlations of site and soil properties were quite similar to the correlations at all observation points (S1 Fig).

The multivariate test of deviance indicated a significant effect of site on *D. dyeri* seedling abundance (p = 0.001) and no significant effect on *D. dyeri* presence (p = 0.065). Both the abundance and presence of *D. dyeri* seedlings were positively correlated with the site ME association, with coefficients (Wald value) of 2.5 and 1.98 respectively.

Seedlings were more likely to be present in darker sites (Table 2, Fig 2, S1 Fig). The association of *D. dyeri* seedling with lower CO was also indicated by the mean difference in CO

**Table 2. Results for four generalized linear models which predict presence and abundance of *D. dyeri* seedlings per observation point at both sites (models 1 and 2), and at the mixed evergreen (models 3 and 4) site only.** ME: Mixed evergreen, MB: Mixed bamboo, BA: Stand basal area, PD.A: Dominance of adult *D. dyeri*.

| Model | Parameter | *Estimate* | Std.Dev./ Std. Error | p | AIC | Deviance explain by the model |
|---|---|---|---|---|---|---|
| (1) Presence of seedlings | Intercept | 3.653 | 1.107 | 0.001 | 153.6 | 11.2% |
| | BA | -0.028 | 0.016 | 0.069 | | |
| | PD.A | 4.358 | 1.584 | 0.006 | | |
| | openness | -0.171 | 0.054 | 0.001 | | |
| | Silt | -0.035 | 0.02 | 0.084 | | |
| (2) Abundance of seedlings | Intercept | 5.33 | 2.109 | 0.0115 | 227.7 | 28.84% |
| | BA | 0.025 | 0.007 | 0.0004 | Std. Err.: 25.8 | |
| | pH | -1.52 | 0.57 | 0.008 | Theta: 19.6 | |
| (3) Presence of seedling on ME site | Intercept | 5.978 | 1.498 | 0.000 | 113.1 | 35.3% |
| | BA | -0.083 | 0.025 | 0.001 | | |
| | pH | -1.817 | 1.227 | 0.138 | | |
| | PD.A | 7.781 | 2.244 | 0.0005 | | |
| | openness | -0.207 | 0.077 | 0.007 | | |
| | Silt | -0.07 | 0.027 | 0.01 | | |
| (4) Abundance of seedlings on ME site | Intercept | 4.597 | 2.359 | 0.0514 | 291.6 | 8.4% |
| | PD.A | 0.029 | 0.01 | 0.0295 | Std. Err.: 0.92 | |
| | pH | -1.302 | 0.6323 | 0.0395 | Theta: 2.32 | |
| (5) Presence of seedlings on MB site | Intercept | -93.448 | 69.728 | 0.18 | 32.384 | 8.0% |
| | Silt | 0.301 | 0.225 | 0.181 | | |
| | pH | 22.747 | 17.083 | 0.183 | | |

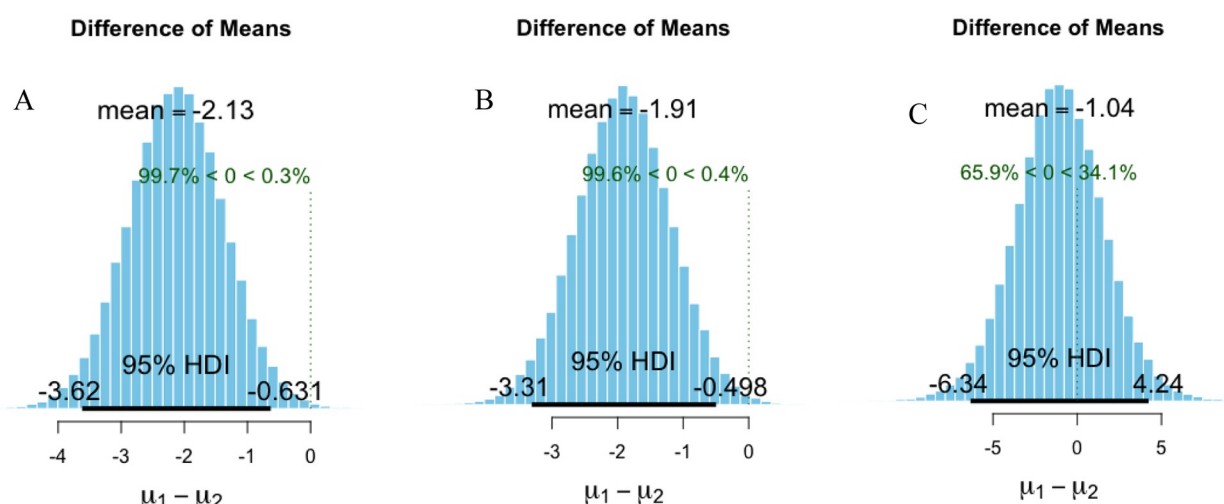

**Fig 2. Difference of means of canopy openness (%) between presence (µ1) and absence (µ2) of seedlings for all data (A), for points on mixed evergreen site (B) and for points on bamboo dominant site (C).**

between the presence and absence observation points of -2.1% (95% credible interval from -3.6 to -0.65) (Fig 2A). The lower CO was also found on ME site with the mean difference of CO between presence and absence points was -1.9% (95% credible interval from -3.3 to -0.48) (Fig 2B). However, the lower CO points where *D. dyeri* seedlings were absent on the MB site was not clear, with mean of difference with the present points was -1.1 (95% credible interval from -6.4 to 4.1) (Fig 2C). The results of the GLM also indicated that presence of seedlings significantly negatively correlated with CO on both sites (p = 0.01), and with ME site (p = 0.007), but not correlated with the MB site (Table 2).

The dominance of adult *D. dyeri* trees (PD.A) was positively correlated with the presence of seedlings for both sites (p = 0.006). PD.A was positively correlated with presence and abundance of seedling on ME site, but was not correlated to these variables on the MB site (Table 2). The results from BMA showed that, PD.A was the fourth most important variable in the presence of seedling model with a probability of 46% coefficient is not equal to zero, coefficient of 1.704 (± 2.218), but PD.A was the least important variable in the abundance model with only 7.4% probability that coefficient is not equal to zero, coefficient of -0.018 (± 0.176).

Seedling abundance was not correlated with CO (Table 2) but it was marginally negatively correlated with stand total basal area (BA) with p = 0.069 (Table 2, but when seedlings were present their abundance was positively correlated with stand total basal area (p = 0.0004, Table 2). The correlations of BA with seedling presence and abundance were significant on the ME site, with a negative correlation with presence, and positive correlation with abundance. However, all of the variables included in the model-averaged analysis were not always negatively or positively correlated with abundance of seedlings (S2 Table). Comparing with other factors, BA showed weaker effects to seedling presence than CO and weaker effects than soil pH to the seedling abundance (Table 2).

When analysed across both sites, seedling abundance was negatively correlated with soil pH and positively correlated with BA (Table 2). In contrast, when the data for the ME site was analysed separately, there was a significant negative correlation between abundance and PD.A (p = 0.03) and also with soil pH (p = 0.04) (Table 2). This consistent negative correlation of pH with abundance of seedlings is reflected in a probability of 95% coefficient of pH not equal to zero out of 97% all the BMA models with coefficient of -1.619 (± 0.629) (S2 Table). In the set

of models for abundance of seedlings on both sites, pH was included in less than 50% of the models, and inconsistently negatively or positively correlated with coefficient of -0.529 (± 0.677). Overall, at both sites soil pH was low (typically less than 5.0) (Table 1). Topsoil silt content had a weakest negative correlation (coefficient = -0.07, p = 0.01) with seedling presence only on the ME site (Table 2).

## 3.2. Shade house study: Seedlings response to light and soil treatments

After 10 months, the seedlings grown in the lowest light level had significantly lower survival rates compared with those at higher light levels ($\chi$-squared = 83.51, df = 5, p < 0.00). While most of the seedlings at medium and high light levels survived (99% survival, only one seedling died in F.M treatment), only 19.4% of seedlings at the lowest light level survived (Table 3). The effect of soil type on seedling survival was evident at the lowest light level with 25% survival of seedlings on degraded soil compared to only 14% seedling survival on forest soil (Table 3). At the highest light level, soil type also appeared to have an effect on seedling health with a lower rate of seedlings with leaf damage (holes or marks by fungal disease or insects) on the forest soil (16.7%) compared to degraded soil (25%).

ANOVA results showed that light level, soil and light: soil interaction significantly influenced seedling height (p = 0.03; Fig 3), however, only light level resulted in a significant difference in seedling diameter (p = 0.000; Fig 3). The seedlings grown at a high light level grew higher (Low–High = -37.1, p = 0.000; Medium—High = -26.35, p = 0.000, Medium–Low = 10.75, p = 0.001) and larger (Low–High = -1.88, p = 0.000; Medium—High = -2.06, p = 0.000, Medium–Low = -0.19, p = 0.73) than those in low light levels (Fig 3). Soils significantly influenced the seedling height at the high light level treatment (p = 0.000; Fig 3A). The seedlings grown on forest soil had significantly higher stem height (Forest—Degraded = 6.39, p = 0.000; Fig 3A) and slightly larger diameters than those grown in degraded soil (Forest—Degraded = 0.15, p = 0.29; Fig 3B) and. The seedlings grown at low and medium light levels were very similar in growth with no difference between the two soils.

The data on seedling leaf characteristics after 10 months was collected from shade house component of three light levels and two soil type. Leaf morphology and anatomy were significantly different among light and soil treatments (Fig 4). Seedlings in higher light and on forest soil had greater leaves of leaf length (p < 0.001) and leaf width (p <0.001) (Fig 4A and 4B) but thinner leaf thickness (Fig 4C), except at the medium light level, leaves of seedlings on forest soil and degraded soils were not significant in length and wide. The leaves of seedlings on forested soil were thinner than those on degraded soil (p < 0.001). Leaf stomatal density was positively correlated with light levels, and seedlings on degraded soil treatments had higher leaf stomatal density compared to those on forest soil treatments (Fig 4D). Leaf chlorophyll

**Table 3. Seedling survival and health status depending on soils and light levels.**

| Treatment | Survival | | Health status (leaf damage) | |
|---|---|---|---|---|
| | Seedling | Rate (%) | Seedling | Rate (%) |
| F.H | 72 | 100 | 12 | 16.7 |
| D.H | 72 | 100 | 18 | 25.0 |
| F.M | 71 | 98.6 | 19 | 26.4 |
| D.M | 72 | 100 | 18 | 25.0 |
| F.L | 10 | 13.9 | 10 | 100 |
| D.L | 18 | 25.0 | 18 | 100 |

F: Forested soil, D: Degraded soil, H: High light level, M: Medium light level, L: Low light level.

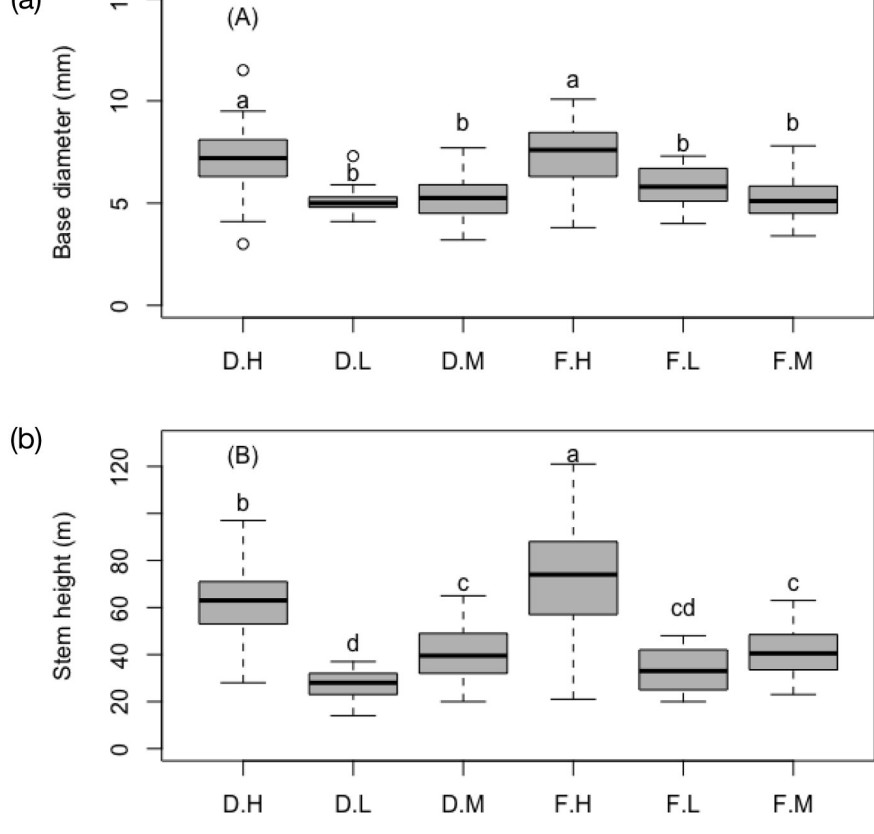

**Fig 3. Seedlings growth in base diameter (A), and height (B).**

content was negatively correlated to light level (Fig 4E). Soil type had a significant effect on chlorophyll content at both low and high light levels but not significant at the medium light level. At the low light level, seedlings on forest soil had lower leaf chlorophyll content than those on degraded soil and the opposite result was shown at high light level.

## 4. Discussion

*Dipterocarpus dyeri* seedlings were more likely to be present at sites with lower canopy openness (Table 1, Table 2, Fig 2). Previous studies have shown that *D. dyeri* is a typical climax tropical shade-tolerant species in Dong Nai [42,58] and this is supported by our results. In our study, the seedlings that survived in low light level conditions had higher leaf chlorophyll content and stomata density (Fig 4) indicating that *D. dyeri* has high light capture efficiency at low light availability [85]. In addition, *D. dyeri* is a large-seeded species (S2B Fig), with recalcitrant seeds [54], a further indicator that it is a shade tolerant tropical climax species [86]. Seedlings which have high leaf area, such as *D. dyeri*, are more likely disadvantaged in large gaps because of dryer soil, higher temperature and greater water loss through their leaves [87]. The *D. dyeri* seedlings growing in the forest understory tended to have a lateral branching trait (S2C Fig), which allows maximum light capture in shade [88]. Our finding that *D. dyeri* seedlings occurred in sites with low light is consistent with previous research that concluded that the late-successional tropical seedlings in the *Dipterocarpus* genus are the most shade-tolerant genus in the Dipterocarpaceae family [42,89–91].

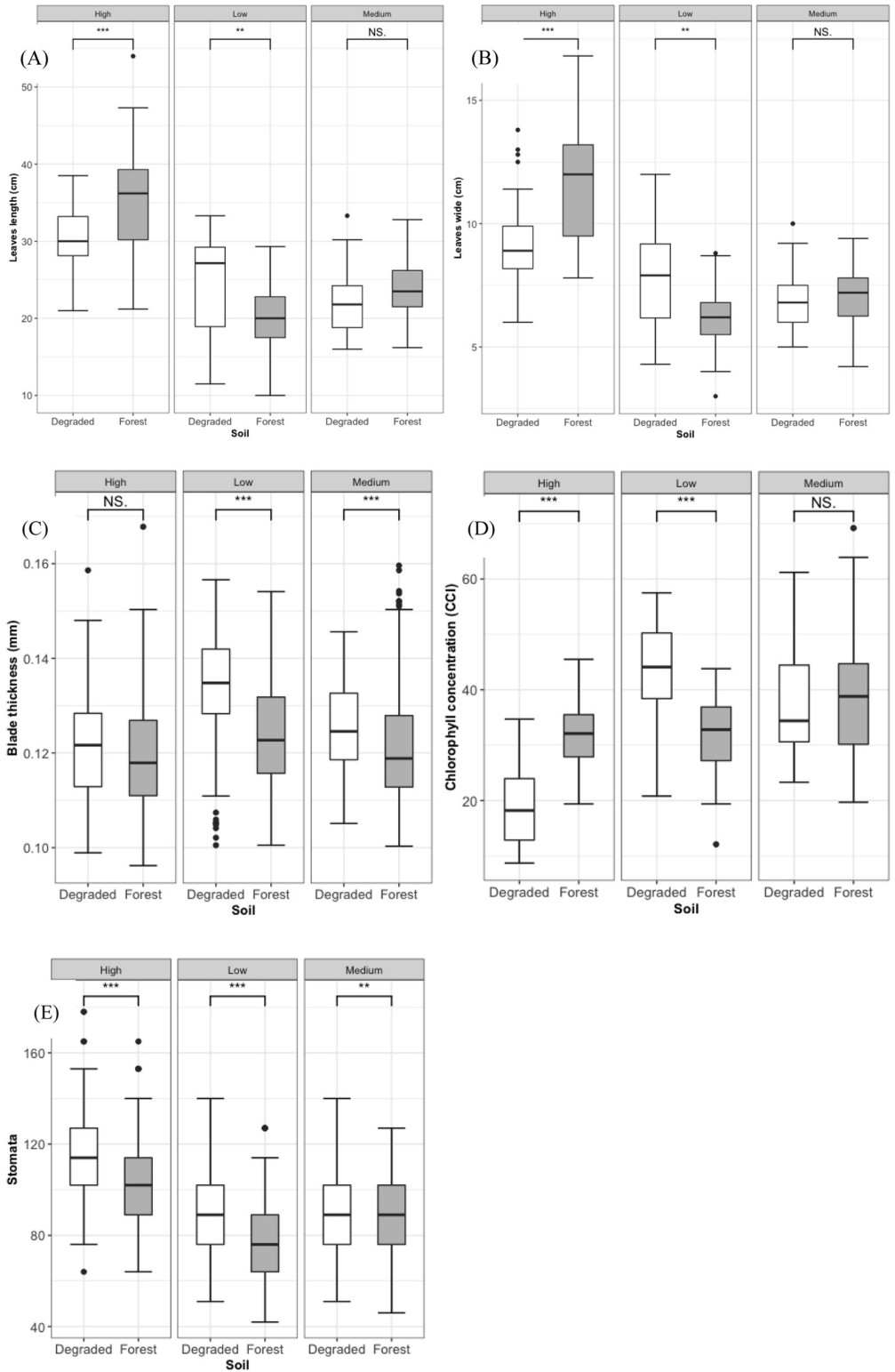

**Fig 4. Influence of soils and light levels to leaves length (A), leaves wide (B) blade thickness (C); chlorophyll concentration (D); and stomatal density (E).**

*Dipterocarpus dyeri* appears to be a broadly adapted generalist dipterocarp species that can survive and grow in a broad range of light and soil conditions. The negative correlation between seedling presence and light was at odds with species abundance findings (which indicated a positive association between abundance and light) (Table 2). This result may be due to the limited range of light availability levels at our study sites (9.5% to 22%). These levels fall within the range appropriate for many shade-tolerant dipterocarps [92–94]. In the shade house, survival and growth of *D. dyeri* was higher at the higher light levels, with only 25% of seedlings surviving at the lowest light level (5%) (Table 3). These results suggest that *D. dyeri* probably can persist at canopy openness lower than 10%, and that it is a generalist species rather than a shade-tolerant specialist. This is consistent with the findings of Denslow [11] and Paoli et al. [94] who found that in rainforest, most canopy species are neither open environment pioneer species nor shade-preferring "climax" forest species but generalists that can tolerate low light but require more open condition to grow into canopy trees [95–97].

*Dipterocarpus dyeri* seedling presence had a weak negative correlation with total forest basal area (BA) (Table 2). However, at the subset of sites where the seedlings were present, their abundance was positively correlated with BA (Table 2). On the surface, this results seems counterintuitive, because typically sites with higher BA are also darker. However, in this study canopy openness was not correlated with BA (S1 Fig) because the sampling time was in the dry season, after deciduous species had dropped their leaves. Seedlings were less likely to be present at higher BA sites because these sites were characterised by a higher deciduous component, which was typically composed of *Cratoxylum* and *Lagerstroemia* species. *Cratoxylum* and *Lagerstroemia* are strongly competitive, long-lived pioneer species and, *D. dyeri* seedling survival is likely to be lower at these sites.

Seedling presence and abundance were strongly positively correlated with the dominance of mother trees (PD.A) (Table 2). This result was expected for a number of reasons. Firstly, the higher PD.A indicated that the site conditions were suitable for the species. Secondly, higher PD.A suggested a more stable seed source in the recovered forest [42,98,99]. Thirdly, *D. dyeri* is a brevideciduous species with leaf-flushing within one week (this study and [42]). It can be grouped with evergreen species similar to two other *Dipterocarpus* species in Thailand [72], so where it dominates basal area, the canopy in the dry season is less open (cf. deciduous forests), and lower light availability was associated with the presence of *D. dyeri* seedlings in our study (Fig 2). This positive correlation between presence/abundance of seedlings and mother trees contradicts previous research describing a negative effect of conspecific adult effects (due to impacts of pathogens and leaf litter) on the survival of early-stage seedlings, which was considered to be a partial explanation for the species richness in tropical forests [100–103]. However, conspecific negative density effects have often been found to be a weaker factor compared to environmental conditions [47,104,105]. Conspecific density affects differently on different species, and at different stages of forest development [106]. For example, *D. dyeri* species produces an abundance of large seeds and large seed mass can be one of the strongest factors compensating for the conspecific negative density effect [107]. In Dong Nai, *D. dyeri* had been logged [58,108], resulting in the abundance of adult *D. dyeri* present today. Hence the density of adult *D. dyeri* trees might not be high enough to show the negative effects.

*Dipterocarpus dyeri* seedling abundance was negatively correlated with soil pH (Table 2). Overall, at all sites, soil pH was low (pH$_{(KCl)}$ from 3.6 to 4.0 or pH$_{(water)}$ of 4.5 to 5.0) (Table 1, S2 Table). Commonly, climax lowland tropical species show a high tolerance to the infertile, acidic and high aluminium concentration of tropical soils (Acrisol and Ferrasols) [20], such as occurs in dipterocarp forests. However, tropical soils can be very spatially variable and while forest soils are generally of low fertility, the higher fertility soils can be found on sloping sites where the soils are younger and have undergone less leaching [20]. The negative correlation

between seedling abundance and pH is attributed to the higher tolerance of evergreen species to strongly acidic soils, compared with deciduous species, which usually prefer higher soil pH (from 5.8 to 6.2) [20]. Evergreen species are also less tolerant of low moisture than deciduous species, as occurs in these sites during the dry season. Lower pH affects the availability (or is correlated with the availability) of a range of beneficial as well as toxic minerals (aluminium, iron, manganese and phosphorus) and these are likely to be the driving factor [20]. The negative correlation between seedling abundance and soil pH was in contrast with findings for *D. sublamellatus* in Borneo, which was not associated with pH [94]. However, in that study the species is considered to be a soil nutrient generalist and therefore would be expected to be more tolerant of a wide range of conditions [94].

Seedling mortality was greatest in the lowest light level in the shade house trial (Table 3). The results show opposite trends in natural vs simulated (shade house) conditions; it is possible that factors other than light might have discernible influence on the presence or absence of seedlings. Local site temporal factors such as litter thickness, soil $CO_2$ concentration, presence of herbivores and pathogens can affect germination and process of seedling establishment. In our case, there was the evidence of *Fusarium* wilt disease (S2D and S2E Fig), which infected the plants in the lower light and higher humidity conditions. In addition, while all seedlings were given the same amount of irrigation the soils in heavy shade appeared to stay wetter. This result was similar to the lower survival of a range of tropical species under lower light availability reported by Augspurger et al. [102] and Gaviria and Engelbrecht [109], and high mortality for *Dipterocarpus* seedlings at low light levels [94,110]. There are other possible and related explanations for high mortality in low light. Firstly, seedling photosynthesis was low and leading to seedlings being less able to resist disease and insect damage. Secondly, saturation of the soils could negatively affect survival of seedlings. Under the low light levels, the photosynthesis rate was low, meaning that less water was lost through leaves and evaporation from soil. Consequently, the soil, which was a Chromic Acrisol with a relatively high clay content maintained high moisture levels [65,111]. The Fe/Al-rich, acidic tropical soil may have become anoxic and therefore toxic for seedlings [112]. The heaviness of this soil may also explain the higher survival rate of seedlings on the lower clay content (degraded soil) at the same light level treatment (Table 3). Our survey also showed that adult *D. dyeri* were found at higher densities on the steep sites (cluster RR1 with 80% total basal area was made of *D. dyeri*) where the dry seasons were relatively wetter and have a shorter time for which they are waterlogged during the rainy season [17,113,114]. Thirdly, *D. dyeri* may be maladapted to flooding and waterlogged soils, as the species shows some morphological characteristics indicative of drought tolerance. For example, *D. dyeri* seedlings have hairy leaves and deep tap roots possibly giving them more tolerance to droughts [115–117].

## 5. Conclusions

Regeneration of *D. dyeri*, the most dominant dipterocarp species in the region, is the key for restoring the forest landscape. *D. dyeri* appears to be a broadly adapted generalist dipterocarp species that can survive and grow in a broad range of light and soil conditions. *D. dyeri* seedlings also survived better where soils had low pH, and good drainage/dry soils rather than wet soils. The strong correlation of *D. dyeri* seedling presence and abundance with the dominance of mother trees is linked to the recovering status of the forest in Dong Nai. At the landscape scale, sites characterised by mixed bamboo (lower available soil water and light level [118– 120]) and shallow soils (on schist and shale soil parent materials) have lower presence and abundance of *D. dyeri* seedlings, indicating that such sites are less suitable for *D. dyeri*; reforestation of these sites should focus on other species. Microsite conditions, especially topographic

location, impact on soil water regime in seasonally dry areas like Dong Nai, and this contributes to the observed distribution patterns of seedlings. With regard to restoration techniques, our findings suggest that (a) nurse crops are not necessary for *D. dyeri* establishment, if the site is moist enough, but (b) nurse crops should be used at drought-prone sites; (c) under the most common tree cover conditions, including intermediate recovered forest or dense plantation, light liberation will be necessary for native forest restoration; (d) restoration should not be attempted on waterlogged sites and (e) using phosphorus or lime carefully to increase soil phosphorus (but maintaining pH < 5.5) should be considered.

## Supporting information

**S1 Data. Data collection methods.**
(DOCX)

**S2 Data. Data of site and soil characteristics at each observation point.**
(DOCX)

**S1 Table. Characteristics of soil used in shade house experiment.**
(DOCX)

**S2 Table. Model-averaged coefficients for site and soil effect on *D. dyeri* seedling presence and abundance.** Values presented are means across each Bayesian model averaging model set. For each parameter, $p \neq 0$ is the probability that the coefficient is not equal to zero.
(DOCX)

**S1 Fig. Correlation of soil and site properties at both sites (A), at ME only (B), at MB only (C), and at observation points where the seedling presented only (D).** Positive correlations are displayed in grey and negative correlations in black. The size of the circle are proportional of the correlation coefficients with the significant level 0.05.
(DOCX)

**S2 Fig. Germinating seeds and seedlings when transplanted in plastic pots (A) Fruit size (B), Branching of under-canopy seedling (C), Seedling died in shade house (D) and Fusarium samples (E).**
(DOCX)

## Acknowledgments

The authors thank Quang-Trung Nguyen and Anh-Linh Nguyen of DNBR for their supports of collecting data in the forest and looking after the shade house experiment.

## Author Contributions

**Conceptualization:** Ha T. T. Do, John C. Grant, Heidi C. Zimmer, J. Doland Nichols.

**Data curation:** Ha T. T. Do.

**Formal analysis:** Ha T. T. Do.

**Funding acquisition:** J. Doland Nichols.

**Investigation:** Ha T. T. Do, Bon N. Trinh.

**Methodology:** Ha T. T. Do, John C. Grant, Heidi C. Zimmer, J. Doland Nichols.

**Project administration:** Ha T. T. Do.

**Resources:** Bon N. Trinh.

**Supervision:** John C. Grant, Heidi C. Zimmer, Bon N. Trinh, J. Doland Nichols.

**Visualization:** Ha T. T. Do.

**Writing – original draft:** Ha T. T. Do, John C. Grant, Heidi C. Zimmer, J. Doland Nichols.

**Writing – review & editing:** Ha T. T. Do, John C. Grant, Heidi C. Zimmer, J. Doland Nichols.

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
