## [Decision Letter · Decision Letter 0]

21 Feb 2020

PONE-D-19-35237

Site conditions for regeneration of climax species, the key for restoring moist deciduous tropical forest in Southern Vietnam

PLOS ONE

Dear Ms Do,

Thank you for submitting your manuscript to PLOS ONE. After careful consideration, we feel that it has merit but does not fully meet PLOS ONE’s publication criteria as it currently stands. Therefore, we invite you to submit a revised version of the manuscript that addresses the points raised during the review process.

ACADEMIC EDITOR: Please revise your manuscript considering all the comments/suggestions of both the reviewers.

We would appreciate receiving your revised manuscript by Apr 06 2020 11:59PM. To enhance the reproducibility of your results, we recommend that if applicable you deposit your laboratory protocols in protocols.io, where a protocol can be assigned its own identifier (DOI) such that it can be cited independently in the future. For instructions see: http://journals.plos.org/plosone/s/submission-guidelines#loc-laboratory-protocols

We look forward to receiving your revised manuscript.

Kind regards,

Arun Jyoti Nath

Academic Editor

PLOS ONE

Journal Requirements:

2. Our internal editors have looked over your manuscript and determined that it is within the scope of our Biodiversity Conservation Call for Papers. This collection of papers is headed by a team of Guest Editors for PLOS ONE (https://collections.plos.org/s/biodiversity). The Collection will encompass a diverse range of research articles on biodiversity conservation, including habitat restoration. Additional information can be found on our announcement page: https://collections.plos.org/s/biodiversity

If you would like your manuscript to be considered for this collection, please let us know in your cover letter and we will ensure that your paper is treated as if you were responding to this call. If you would prefer to remove your manuscript from collection consideration, please specify this in the cover letter."

3. In your Methods section, please provide additional location information of the study area, including geographic coordinates for the data set if available.

4. In your Methods section, please provide additional information regarding the permits you obtained for the work. Please ensure you have included the full name of the authority that approved the field site access and, if no permits were required, a brief statement explaining why.

a)    You may seek permission from the original copyright holder of Figure 1 to publish the content specifically under the CC BY 4.0 license.  

Reviewers' comments:

Reviewer's Responses to Questions

**Comments to the Author**

1. Is the manuscript technically sound, and do the data support the conclusions?

Reviewer #1: Yes

Reviewer #2: Yes

2. Has the statistical analysis been performed appropriately and rigorously? 

Reviewer #1: Yes

Reviewer #2: Yes

3. Have the authors made all data underlying the findings in their manuscript fully available?

Reviewer #1: Yes

Reviewer #2: Yes

4. Is the manuscript presented in an intelligible fashion and written in standard English?

Reviewer #1: Yes

Reviewer #2: Yes

5. Review Comments to the Author

Reviewer #1: The study on Site conditions for regeneration of climax species, the key for restoring moist deciduous tropical forest in Southern Vietnam has determined the primary ecological processes influencing Dipterocarpus dyeri seedling establishment and survival. The study aimed to improve understanding the secondary forest regeneration requirements of an ecologically important tree seedling. The findings of this study will be very much helpful to execute silvicultural practices for planting stock preparation and degraded forest restoration. The experimental results have been analysed using appropriate statistical tools and techniques and presented well. I have some queries as listed below:

1. The soil properties of both the sites were very close to each others. However, sulphur content in degraded soil is quite higher than the forest soil. This needs to be explained.

2. Table-3 Seedling survival rates in low light level were less than the medium and higher light levels. However, at low light level and both the soil type all the seedlings were infected by fungus. The observations indicate that soil properties have no influence in seedling health irrespective of differences in nutrients in healthy and degraded forest soils, only light intensity causes fungal infection. Survival rates of seedlings in degraded and forest soil varied significantly but the health status was same in both the soil types. Besides, in natural forests seedling abundance was not correlated with canopy openness. So, in natural forest light intensity have no influence on seedling growth but in shade house experiments light level played a crucial role in seedling health and survival. This need to be justified properly.

3. Authors have mentioned that D. dyeri is a shade tolerant tree species while the shade house experimental results revealed better performance i.e. seedling survival rate and health in higher light levels. This may also be explained properly. Because at high and medium light level soil properties could not play any role in survival as 100 % seedlings were survived. If the tree is shade tolerant and seedling abundance was more under dense canopy the survival rate of seedlings should be more in low light levels than high level.

4. Page 21 line 277 please correct leaf wide as leaf width.

The manuscript may be considered for publication after minor revision.

Reviewer #2: The authors of this paper have investigated the presence and abundance of Dipterocarpus dyeri seedlings across a range of environmental conditions, and performed an experiment in shade house to evaluate the effect of light levels on the mortality and health status of the seedlings of this species. The study is nevertheless interesting and is a necessary research work. The authors presented a reasonable methodology and results, and the paper in general is well written. However, there are a lot of typo errors and jumping off words, besides, some clarity on data collection and site information is missing. I suggest an acceptance after incorporation of these in the revised manuscript.

Specific comments:

L22. Abstract. Provide the level of significance

L25. Mention the % of mortality

L30-32. Sentences not clear. Briefly mention why recovery action is needed while D. dyeri is already a dominant species

L33. Provide keywords after the abstract.

L 85-86. Sentences not clear.

L99. Please give authority of the species (Dipterocarpus dyeri) used for the first time.

L106. Add a paragraph mentioning the treatment level of the species in South East Vietnam and causal factors associated to it

L110. Please provide the geo coordinates of the study sites

L117. 44 to 130 m asl???

L137. The authority of the species may be deleted

L146-154. What was the proximity between two sites (MB & ME sites)? How many 400m2 plots were established? What was the size of the other three plots (where no seedling of the target species, but the adult species found). Is it the same with as sampling clusters? Were there other plots too where no seedling got established? How such plots were compared? How many plots were considered for vegetation survey and for other site characteristics? Please mention.

L156. The methods of measurement of different variables that are included in the table-1 should be briefly included under the head ‘Data collection”.

L162-165. Mention how light levels were quantified. What was the experimental design and how many replicates were used for each treatment. How the seedlings were grown (in polypot???) and quantity soil added, and for how long the seedlings were allowed to grow in the shade house, need to be included.

L182. How canopy openness was measured while sampling was carried out in dry season where deciduous associates of D. dyeri shed their leaves?

L218 Expand the abbreviations used in Table 2 or please add the key as in table 1. The significant codes 0.05 ‘.’, 0.1 ‘ ‘ should be deleted as no such code is mentioned in the table.

L254 Provide seedling survival percent at medium and high light levels

L262 Table 3 Leaf damage may be put in parentheses after Health status for better clarity

L263. Expand all the abbreviations used as treatments in table 3.

L263. LSD value may be included against each column in Table 3.

L268. Influence to the seedlings be changed to influenced the seedlings

L269-270. Seedling grown in forested soil had slightly larger diameter? Compared to what?

Line 273. Based diameter should be replaced with base diameter

L276-279. The sentences may be rewritten for better clarity. Forest soil had greater leaves of leaf length??? Leaf wide may be replaced with leaf width.

L280-283. First time, results on leaf stomatal density and chlorophyll are mentioned. Please mention these in data collection under shade house component.

L298-299. May not necessarily be true always. Eg. Teak a large leaved species is a light demander.

L309. The results shows opposite trend in natural vs simulated (shade house) conditions; there may be possibilities that other factors (other than light) might have discernable influence on the presence or absence of seedling. The factors like litter thickness, soil CO2 concentration too would affect germination and process of seedling establishment. Please include these in the discussion.

L349. Typo errors. Please rectify the errors.

L363. Please specify if there was irrigation in the shade house as the level of irrigation would affect humidity, the result however would be different if there was controlled irrigation in the shade house

L381-388. This portion may be shifted to the conclusion part of the paper.

L390-394. These sentences may be shifted to the discussion part.

Fig 3. based diameter be changed to base diameter

Fig 4. leaves wide be replaced with leaves width

6. PLOS authors have the option to publish the peer review history of their article (what does this mean?). If published, this will include your full peer review and any attached files.

Reviewer #1: No

Reviewer #2: No

---

## [Author Response · Author response to Decision Letter 0]

14 Apr 2020

Specific comments:

Abstract. Provide the level of significance

L22. We added p values “Seedling presence (p = 0.065) and abundance varied significantly (p = 0.001) between the two sites,”

L25. Mention the % of mortality

L26-27. We add that “Seedling survival was significantly lower at the lowest light level (<10% full irradiance) at 13% for the forest soil and 25% for degraded soil. At higher irradiance the seedling survival rates were greater than 99%.”

L30-32. Sentences not clear. Briefly mention why recovery action is needed while D. dyeri is already a dominant species

L31-33. We have reworded and clarified the following sentences: “Historically, Dipterocapus dyeri was dominant in moist deciduous tropical forest across south-eastern Vietnam, but today it is rare. Active management of these recovering forests is essential in order to recover this high-value, climax forest species.

L33. Provide keywords after the abstract.

L35-36. We add “Keywords: canopy openness, Dipterocarpus dyeri Pierre, microsite, restoration, seasonally dry tropical forests, secondary forest”

L 85-86. Sentences not clear. 

L95. We have reworded and clarified the following sentence: “Dipterocarpaceae was previously abundant throughout southeast Asia, however most remaining forests are heavily disturbed and are dominated by deciduous tree species and bamboo”

L99. Please give authority of the species (Dipterocarpus dyeri) used for the first time.

L115. We have added authority of the species: “Dipterocarpus dyeri Pierre”

L106. Add a paragraph mentioning the treatment level of the species in South East Vietnam and causal factors associated to it

L105-112. We add one paragraph: “Dipterocarpus dyeri is native to Myanmar, Thailand, Viet Nam, Cambodia and Peninsular Malaysia. The species has been globally assessed as Endangered in the IUCN Red list [53,69]. Most dipterocarp species, including Dipterocarpus dyeri Pierre, are high value timber species, and as such, have been selectively logged across South-east Asia [42]. Dipterocarpus dyeri has been targeted in particular because of its high density wood (~ 0.8 g/cm3) and also their resins (dammar) which can be extracted for caulking boats, varnish paint, and medicine [42,53–56]. Dipterocarpus dyeri today is one of the rarest species in the forest and there is limited information on species ecology regeneration and growing site conditions [53,57].” 

L110. Please provide the geo coordinates of the study sites

L128-129. We have added coordinates of the Dong Nai Biosphere Reserve: “The Reserve extends between 11°20’50”N – 11°50’20”N and 107°09’05”E – 107°35’20”E”.

L117. 44 to 130 m asl???

L136. We have corrected this to: “130 m asl”.

L137. The authority of the species may be deleted

L156. We have deleted species authority. 

What was the proximity between two sites (MB & ME sites)? 

L167. We added “ approximately 20 km from ME site” 

How many 400m2 plots were established? 

L 165. We added: “from 18 sampling clusters (represented by plot data, Table 1, Fig 1, S1)”.

What was the size of the other three plots (where no seedling of the target species, but the adult species found). Is it the same with as sampling clusters? 

L191. We added: “If none of seedlings was found, then a similar 400 m2 plot was established”.

Were there other plots too where no seedling got established? 

L171-173. We added “If no seedlings were found, then a similar 400 m2 plot was established for surveying site and soil conditions, with a mother tree as the centre and five observation points (each of the four corners and at the centre of the plot)”.

How such plots were compared? How many plots were considered for vegetation survey and for other site characteristics? Please mention. 

L164-165. “A total of 122 observation points (2 m x 2 m) were taken from 18 sampling clusters (represented by plot vegetation data, Table 1, Fig 1, S1).”

L156. The methods of measurement of different variables that are included in the table-1 should be briefly included under the head ‘Data collection”.

L174-197. We add method for data collection from forest of vegetation composition, light level and environment:

“Vegetation composition. In each plot, all woody species (excluding lianas) were recorded and divided into three classes based on the diameter at 1.3 m height from the ground (DBH): adults (DBH � 10 cm), juveniles (5 cm � DBH < 10 cm). Adults and juveniles were surveyed from 400 m2 plots while seedlings of all species were surveyed from the observation points.

Soil. At each observation point, a description of soil from 0-35 cm depth was made, including horizons, colour and texture. One topsoil (0-10 cm depth) sample which was a mixture of soil from 3-5 observation points was taken for soil chemical characterisation of each sampling cluster. At each sampling cluster one core sample (5 cm diameter x 5 cm length) was taken for topsoil bulk density calculation. 

“Light. A hemispherical canopy photograph was taken at each observation point at 170 cm above the ground. We chose this height to avoid the shrub and herb layers. These photos were used to determine canopy openness (%) using Gap Light Analyzer (GLA) software [1], after processing the image through SideLook [2] using automatic thresholding. The procedure of using automatic threshold to adjudge colour canopy photographs is recommended for assessing canopy openness [3].”

Environment. Basic site conditions were described at each observation point, including slope (degrees), aspect (degrees), outcrop rock (% cover), forest floor vegetation cover (%), litter cover (%) and litter thickness (cm). Topographic wetness index (TWI) [4], was calculated as a soil moisture indicator [5,6] for each observation point was derived by Arctoolbox from a digital elevation model (DEM) through SAGAGIS, AcrGIS v 10.1. A DEM of 30 x 30 m grid spacing was downloaded from https://earthexplorer.usgs.gov/.”

L162-165. Mention how light levels were quantified. What was the experimental design and how many replicates were used for each treatment. How the seedlings were grown (in polypot???) and quantity soil added, and for how long the seedlings were allowed to grow in the shade house, need to be included.

L198-212. We have added detail on how the shade house experiment was established, including measurement of light levels:

“Shade house experiment establishment

Seeds were collected from one mother tree from Cay Gui Forest Ranger Station (approximately 500 m from plot CG1). The seedlings were germinated using moist sand for 20-30 days until developing to seedlings with two true leaves (Figure A.2 B) which were then transplanted in plastic pots (8 cm diameter x 40 cm height). There were six treatments of three light levels (high, medium and low) and two sources of soil (old growth forest and heavily logged and degraded forest) with a total of 432 seedlings (72 seedlings per treatment). Each light treatment was set up in a shelter with light levels controlled by shade cloth, one layer for the high level, doubling for the medium level and tripling for the low level. The light levels were confirmed by the measurement of leaf area index (LAI) by using Li-2200. The LAI at the three shelters were 2.9, 5.4 and 7.1 giving estimations of canopy openness for each light levels were proximately 45%, 20% and 5% of full sun, respectively. Soil used for the trial was the topsoil of a Chromic Acrisol (Fp) and it was collected from two sites. Both soils were light clays in texture and the old growth forest soil (F) was higher in organic matter, total nitrogen and CEC than the degraded site soil (D) (Appendix 2).”

L182. How canopy openness was measured while sampling was carried out in dry season where deciduous associates of D. dyeri shed their leaves? 

L186-191. We added light level measurement and analysis canopy openness:

“Light. A hemispherical canopy photograph was taken at each observation point at 170 cm above the ground. We chose this height to avoid the shrub and herb layers. These photos were used to determine canopy openness (%) using Gap Light Analyzer (GLA) software [1], after processing the image through SideLook [2] using automatic thresholding. The procedure of using automatic threshold to adjudge colour canopy photographs is recommended for assessing canopy openness [3].”

Expand the abbreviations used in Table 2 or please add the key as in table 1. The significant codes 0.05 ‘.’, 0.1 ‘ ‘ should be deleted as no such code is mentioned in the table.

L270-271. We have added the abbreviations: “ME: Mixed evergreen, MB: Mixed bamboo, BA: Stand basal area, PD.A: Dominance of adult D. dyeri”

L272. We have deleted the significance codes.

Provide seedling survival percent at medium and high light levels

L308. We added seedling survival percent ”(99% survival, only one seedling died in F.M treatment)”

Table 3 Leaf damage may be put in parentheses after Health status for better clarity

L319 – Table 3. We have corrected this to read: “ Health status (Leaf damage)”

L263. Expand all the abbreviations used as treatments in table 3.

L317. We have added abbreviations: “F: Forested soil, D: degraded soil, H: high light level, M: medium light level, L: Low light level”

L263. LSD value may be included against each column in Table 3.

As table 3 reports the results of a Chi squared test it is not appropriate to include least significant values (these are commonly reported alongside ANOVA tables); however we have added the LSD values associated with the results of the ANOVA, reported in Figure 3 (L322-332): “ANOVA results showed that light level, soil and light: soil interaction significantly influenced seedling height (p = 0.03; Fig 3), however, only light level resulted in a significant difference in seedling diameter (p =0.000; Fig 3). The seedlings grown at a high light level grew higher (Low – High =-37.1, p = 0.000; Medium - High = -26.35, p = 0.000, Medium – Low = 10.75, p =0.001) and larger (Low – High =-1.88, p = 0.000; Medium - High = -2.06, p = 0.000, Medium – Low = -0.19, p =0.73) than those in low light levels (Fig 3). Soils significantly influenced the seedling height at the high light level treatment (p=0.000; Fig 3A). The seedlings grown on forest soil had significantly higher stem height (Forest-Degraded = 6.39, p =0.000; Fig 3A) and slightly larger diameters than those grown in degraded soil (Forest-Degraded =0.15, p=0.29; Fig 3B) and. The seedlings grown at low and medium light levels were very similar in growth with no difference between the two soils.

We have also improved Fig 3 by presenting the results from ANOVA test and LSD tests.

L268. Influence to the seedlings be changed to influenced the seedlings

L328. We have changed this phrase to read: “Soils significantly influenced the seedling height”

L269-270. Seedling grown in forested soil had slightly larger diameter? Compared to what?

L330. We have added: “than those grown in degraded soil” 

Line 273. Based diameter should be replaced with base diameter

L334. We have changed to: “Fig 3. Seedlings growth in base diameter (A), and height (B).” 

L276-279. The sentences may be rewritten for better clarity. Forest soil had greater leaves of leaf length??? Leaf wide may be replaced with leaf width.

L338-339. We have improved the sentence :“Seedlings in higher light and on forest soil had greater leaves of leaf length (p < 0.001) and leaf width (p <0.001) (Fig 4 A, B) but thinner leaf thickness (Fig 4C)”

L280-283. First time, results on leaf stomatal density and chlorophyll are mentioned. Please mention these in data collection under shade house component.

L336-337. We added in the beginning of the paragraph: “The data on seedling leaf characteristics after 10 months was collected from shade house component of three light levels and two soil type” 

L298-299. May not necessarily be true always. Eg. Teak a large leaved species is a light demander.

L363. We changed to “Seedlings which have high leaf area, such as D. dyeri, are more likely disadvantaged in large gaps”.

L309. The results shows opposite trend in natural vs simulated (shade house) conditions; there may be possibilities that other factors (other than light) might have discernable influence on the presence or absence of seedling. The factors like litter thickness, soil CO2 concentration too would affect germination and process of seedling establishment. Please include these in the discussion

L430-437. We added this point to our discussion “ Seedling mortality was greatest in the lowest light level in the shade house trial (Table 3). The results show opposite trends in natural vs simulated (shade house) conditions; it is possible that factors other than light might have discernible influence on the presence or absence of seedlings. Local site temporal factors such as litter thickness, soil CO2 concentration, presence of herbivores and pathogens can affect germination and process of seedling establishment. In our case, there was the evidence of Fusarium wilt disease (S2 Fig 2C), which infected the plants in the lower light and higher humidity conditions. In addition, while all seedlings were given the same amount of irrigation the soils in heavy shade appeared to stay wetter.”

L349. Typo errors. Please rectify the errors.

L417. We have corrected this: “, the higher fertility soils can be found on sloping sites“

L363. Please specify if there was irrigation in the shade house as the level of irrigation would affect humidity, the result however would be different if there was controlled irrigation in the shade house

L436-437. We added “while all seedlings were given the same amount of irrigation but soils in heavy shade appeared to stay wetter”.

L381-388. This portion may be shifted to the conclusion part of the paper.

L468-500. We have moved this section to the conclusion, as suggested: “With regard to restoration techniques, our findings suggest that (a) nurse crops are not necessary for D. dyeri establishment, if the site is moist enough, but (b) nurse crops should be used at drought-prone sites; (c) under the most common tree cover conditions, including intermediate recovered forest or dense plantation, light liberation will be necessary for native forest restoration; (d) restoration should not be attempted on waterlogged sites and (e) using phosphorus or lime carefully to increase soil phosphorus (but maintaining pH < 5.5) should be considered.”

L390-394. These sentences may be shifted to the discussion part.

L379-382. We moved to discussion “This is consistent with the findings of Denslow [11] and Paoli et al. [92] who found that in rainforest, most canopy species are neither open environment pioneer species nor shade-preferring “climax” forest species but generalists that can tolerate low light but require more open condition to grow into canopy trees [93–95].”

---

## [Decision Letter · Decision Letter 1]

7 May 2020

Site conditions for regeneration of climax species, the key for restoring moist deciduous tropical forest in Southern Vietnam

PONE-D-19-35237R1

Dear Dr. Do,

We are pleased to inform you that your manuscript has been judged scientifically suitable for publication and will be formally accepted for publication once it complies with all outstanding technical requirements.

With kind regards,

Arun Jyoti Nath

Academic Editor

PLOS ONE

Additional Editor Comments (optional):

Reviewers' comments:

Reviewer's Responses to Questions

**Comments to the Author**

1. If the authors have adequately addressed your comments raised in a previous round of review and you feel that this manuscript is now acceptable for publication, you may indicate that here to bypass the “Comments to the Author” section, enter your conflict of interest statement in the “Confidential to Editor” section, and submit your "Accept" recommendation.

Reviewer #1: All comments have been addressed

2. Is the manuscript technically sound, and do the data support the conclusions?

Reviewer #1: Yes

3. Has the statistical analysis been performed appropriately and rigorously? 

Reviewer #1: Yes

4. Have the authors made all data underlying the findings in their manuscript fully available?

Reviewer #1: Yes

5. Is the manuscript presented in an intelligible fashion and written in standard English?

Reviewer #1: Yes

6. Review Comments to the Author

Reviewer #1: Authors have adequately addressed the issue raised by the reviewers. The manuscript may be considered for publication in its current form

7. PLOS authors have the option to publish the peer review history of their article (what does this mean?). If published, this will include your full peer review and any attached files.

Reviewer #1: No

---

## [Editor Report · Acceptance letter]

11 May 2020

PONE-D-19-35237R1 

Site conditions for regeneration of climax species, the key for restoring moist deciduous tropical forest in Southern Vietnam 

Dear Dr. Do:

I am pleased to inform you that your manuscript has been deemed suitable for publication in PLOS ONE. Congratulations! Your manuscript is now with our production department. 

With kind regards,

on behalf of

Dr. Arun Jyoti Nath 

Academic Editor

PLOS ONE